# Morphological, Mechanical and Hydrodynamic Aspects of Diaphragmatic Lymphatics

**DOI:** 10.3390/biology11121803

**Published:** 2022-12-12

**Authors:** Daniela Negrini

**Affiliations:** Department of Medicine and Surgery, University of Insubria, 21100 Varese, Italy; daniela.negrini@uninsubria.it; Tel.: +39-0332-397104

**Keywords:** serosal lymphatic drainage, diaphragmatic lymphatic loops, diaphragmatic lymph kinetics, HCN-channels in diaphragmatic lymphatics, transdiaphragmatic fluid flux

## Abstract

**Simple Summary:**

The diaphragm is both the main respiratory muscle and the septum separating the thoracic from the abdominal cavity. As most tissues, it is supplied by a vascular lymphatic system whose function is to absorb fluid, solutes and cells from the interstitial tissue surrounding the cells and to propel the so formed lymph back into the venous blood stream. However, the diaphragmatic lymphatic network plays an additional role, being directly involved in draining fluid from the pleural and peritoneal cavities. The aim of this review is to present a comprehensive description of the structure and function of the diaphragmatic lymphatic network, including its unique morphology, the mechanisms supporting diaphragmatic lymph formation and progression and the importance of the mechanical events developing in the diaphragm during tidal breathing on lymph formation and propulsion. This information may prove to be useful in understanding some abnormal fluid accumulation in the pleural or peritoneal cavities observed in clinical settings.

**Abstract:**

The diaphragmatic lymphatic vascular network has unique anatomical characteristics. Studying the morphology and distribution of the lymphatic network in the mouse diaphragm by fluorescence-immunohistochemistry using LYVE-1 (a lymphatic endothelial marker) revealed LYVE1^+^ structures on both sides of the diaphragm—both in its the muscular and tendinous portion, but with different vessel density and configurations. On the pleural side, most LYVE1^+^ configurations are vessel-like with scanty stomata, while the peritoneal side is characterized by abundant LYVE1^+^ flattened lacy-ladder shaped structures with several stomata-like pores, particularly in the muscular portion. Such a complex, three-dimensional organization is enriched, at the peripheral rim of the muscular diaphragm, with spontaneously contracting lymphatic vessel segments able to prompt contractile waves to adjacent collecting lymphatics. This review aims at describing how the external tissue forces developing in the diaphragm, along with cyclic cardiogenic and respiratory swings, interplay with the spontaneous contraction of lymphatic vessel segments at the peripheral diaphragmatic rim to simultaneously set and modulate lymph flow from the pleural and peritoneal cavities. These details may provide useful in understanding the role of diaphragmatic lymphatics not only in physiological but, more so, in pathophysiological circumstances such as in dialysis, metastasis or infection.

## 1. Introduction

The diaphragm is a unique tissue whose structure is specifically adapted to drive lung expansion and collapse during the respiratory cycle, while also compartmentalizing the thoracic from the abdominal cavity. 

In addition to its main respiratory function, the diaphragmatic tissue plays an important role in setting and maintaining the fluid volume and hydraulic pressures in the pleural and peritoneal cavities. Indeed, the diaphragm is equipped with a well-developed network of lymphatic vessels whose function is to produce and propel diaphragmatic lymph produced by the drainage of fluid, solutes and cells from the surrounding interstitial space: the pleural and the peritoneal cavities. While the tendinous and muscular diaphragm contribute little to total diaphragmatic lymph production, the latter mostly depends upon the drainage of pleural and peritoneal fluid. Therefore, the diaphragmatic lymphatics network plays a crucial role in several functions, including lung-chest-wall mechanical coupling and control of pleural and peritoneal fluid and solute turnover. 

The aim of this review is to present a comprehensive analysis of what is actually known on: (a) the very peculiar morphology of the diaphragmatic lymphatics, (b) the mechanism supporting diaphragmatic lymph formation and progression, (c) the physiological role of the diaphragm in the pleuro-peritoneal fluid dynamics and (d) the relevance of the lymphatic diaphragmatic network in clinical settings.

## 2. Morpho-Functional Description of the Diaphragmatic Lymphatic Network

The existence of mesenteric, inguinal and axillary lymph nodes was described by Hippocrates of Cos (ca. 460 BC–ca. 370 BC) and later by Rufus of Ephesus, who practiced medicine in Imperial Rome in the 2nd century B.C.; however, it was only in the 18th century that Paolo Mascagni (1752–1815), professor of Anatomy in Siena and Florence, realized the first systematic description of the lymphatic system in humans. 

By the end of the 18th century, the gross anatomy of the lymphatic system was mostly known, yet the absorptive nature of lymphatic vessels from the body tissues was identified only in 1762 by William Hunter (1718–1783) who correctly proposed that the lymphatics were not “*continuations from arteries, but a particular system of vessels by themselves*” and that lymph was formed by the absorption of fluid from the tissues into peripheral lymphatics. 

### 2.1. Lymphatic Stomata

The German pathologist, Friedrich Daniel von Recklinghausen (1833–1910), introduced the concept of connective tissue drained by lymphatics, revealed the existence of the Kampfmeier foci and was the first to propose the presence of lymphatic “stomata” on the peritoneal surface of the diaphragm [1]. Since then, very few studies performed a thorough analysis of the stomata shape, size and distribution of pleural and peritoneal stomata, encountered only in the parietal mesothelium covering the inner thoracic surface, the mediastinal surface of the heart, the pleural and peritoneal surface of the diaphragm and the inner abdominal surface. Visceral mesothelia, like the pleura covering the lung surface, are devoid of stomata. The mesothelial layer consists of adjacent mesothelial cells connected through intercellular tight junctions and desmosomes [2,3] that limit paracellular transport of large solutes to the submesothelial interstitium. Within such a compact surface, stomata consist of porous-like discontinuities of the parietal mesothelium (Figure 1) at the confluence between the mesothelial and lymphatic endothelial cells [4,5,6], forming a direct connection to the submesothelial lymphatic structures.

Although documented over the costal and the mediastinal parietal pleura (average density: ~100/cm^2^ [5]), the greater stomata density is encountered in the diaphragm (Table 1) where they are spread over the tendinous (T) and the muscular (M) region of both the pleural (PL) and peritoneal surfaces (PE) [personal communication, Gordon Conference 2012]. The average stomata radius varies among the diaphragmatic regions, being the largest stomata encountered in the PL-T region; in addition, they tend to be elliptical in shape, particularly in PL-T, as indicated by the lower roundness index. The stomata hole presents various configurations (Figure 1): (a) wide open round-to-elliptical holes, (b) open holes but with septa partially gating the entrance or (c) completely occluded holes, suggesting the existence of a potentially gated entrance and/or a sequence of stomata development, from newly formed to completely open configuration. 

Because of their diameter (Table 1 and [4,5]) and unlike what was observed through the continuous visceral mesothelium, stomata offer no resistance to the transit of large molecular size solutes, such as plasma proteins and even cells. 

### 2.2. Lymphatic Lacunae and LLS 

After its beginning with stomata, the diaphragmatic lymphatic network is hierarchically organized into (1) submesothelial lacunae running beneath both the pleural and peritoneal surfaces; (2) small lymphatic vessels running through the skeletal muscles and tendinous fibres; (3) large collecting lymphatics in the deep central stratum of diaphragmatic tissues [6]. Although lymphatic lacunae have been described though optical and electronic microscopy [6,7], a better definition of their shape, location and distribution can be appreciated through a fluorescence-immunohistochemical approach coupled to confocal microscopy analysis [personal communication, Gordon Conference 2012]. Data obtained in whole mounts, frozen cross-sections of the mouse diaphragm using LYVE-1 (a lymphatic endothelial marker) revealed that LYVE-1-stained lymphatic structures were clearly visible on the muscle through the tendinous portions on both the pleural (PL) and the peritoneal sides of the mouse diaphragm [8,9], although with significantly different configurations and distributions (Figure 2). LYVE1^+^ lymphatic structures covered a greater percentage of the peritoneal (≈29.2%) than of the pleural (≈5.3%) diaphragmatic surface. 

### 2.3. Lymphatic Structures of the Tendineous Pleural and Peritoneal Diaphragm

Because of the thinness of the tendinous region in the mouse diaphragm (≈260 µm), its lymphatic network could usually be visualized from both the pleural (PL-T) and the peritoneal (PE-T) sides of the tendinous diaphragmatic region (Figure 2). Linear vessels, a few millimetres long and with an average diameter of ≈50 µm, departed from the transition region where the tendon connects to the muscle. More or less complex interconnected loops, up to ≈600 µm long, formed when a shorter transverse vessel with a diameter of ≈50 µm departed from the main tributary vessels, almost perpendicularly with respect to the major linear vessel direction (Figure 2).

As in other tissues, lymphatics of the diaphragm are equipped with two types of unidirectional valves: (a) primary valves in the vessel wall, formed by cytoplasmic extensions of adjacent endothelial cells protruding into the capillary lumen [6,10], which regulate fluid and solute entrance from the interstitial space into the lymphatic lumen; (b) intraluminal valves, formed by leaflet protruding from the lymphatic vessel wall [11,12], which direct lymph through the vessel network towards the larger collecting lymphatics. These latter valves, also called secondary, segment cylindrical lymphatics into a sequence of lymphangions, the lymphatic functional unit, which were clearly observed in both PL-T and PE-T in linear vessels and in loops [6,13,14,15,16,17]. 

A well-defined, abundant lymphatic mesh was also observed in the muscular diaphragm, but with strikingly different morphological characteristics between PL and PE sides.

### 2.4. Lymphatic Structures of the Muscular Pleural Diaphragm

On the PL-M side, most LYVE1^+^ configurations were constituted by linear lymphatic and loops (Figure 2B). Linear vessels were arranged in a longitudinal direction almost parallel to that of the underlying diaphragmatic muscle fibres. Lateral branches connected adjacent tubular vessels to form complex loops. The diameter of the main tributary vessels (≈30 µm) and the length of the loops (≈300 µm) were smaller than in PL-T. In addition, the density of small tubular lymphatics (Φ ≤ 15 μm) was greater in PL-M than in PL-T, while this distribution was the opposite for large tubular lymphatics (Φ ≥ 15 μm). Unlike what was observed in PL-T, PE-T and PE-M regions, only few clearly observable lymphangions with an average length and diameter of ≈200 µm and 50 µm, respectively, were encountered in PL-M zone. In addition, vessel density and tortuosity were greatly increased (Figure 2B) in the most peripheral diaphragmatic rim, likely corresponding to the thoracic tendinous insertion of skeletal muscle fibres and facing the diaphragm-chest wall apposition zone [15].

Smaller LYVE1^+^ capillaries of an average length and width of ≈70 µm and 8 µm, respectively, were observed running parallel at a reciprocal distance of ≈50 µm between adjacent skeletal muscle fibres. About 70 % of these vessels displayed identifiable intraluminal valves which delimited lymphangions about 400 μm long. Lymphangion-endowed vessels (about 3 mm long) were about three times longer than those with no valves/lymphangions. Perpendicularly oriented vessels were significantly larger and shorter than parallel vessels with lymphangions, but comparable in length with the parallel vessels without valves. Intraluminal valves and lymphangions were found in only ≈7% of the perpendicular analysed vessels [15]. 

### 2.5. Lymphatic Structures of the Muscular Peritoneal Diaphragm

On the PE-M side, an extensive and unique LYVE1^+^ flattened lacy-ladder shaped membranous structure (LLS), consistent with previous descriptions of large submesothelial lacunae [7,15], was observed (Figure 2D). The LLS, never encountered on the PL-M side, could be many millimetres long and up to 1 mm wide, laying over a few to dozens of diaphragmatic muscle fibres (average cross-sectional diameter ≈40 µm). The major axis of the LLS structures was aligned to the underlying diaphragmatic fibres. LLS was not homogeneously arranged along the major axis of muscular diaphragmatic fibres. Indeed, parts of the LLS were organized around “void spaces”, where LYVE1^+^ lymphatic endothelial cells were not found, of variable shape and dimensions. The LLS covered a very significant fraction (≈15%) of the PE-M surface area and occupied ≈34% of the PE-M tissue volume, respectively. No lymphangions were observed in LLS. 

Stomata observed in PL-T, PL-M, PE-T and PE-M regions of the diaphragm were invariably located over the surface of underlying lymphatic vessels or LLS (Figure 1). The stomata density per unit surface area of underlying lymphatic structure was rather uniform on the tendinous and muscular sector of both the pleural and peritoneal sides but was significantly higher (Table 1) in PE-T and PE-M compared to the corresponding PL-T and PL-M regions [personal communication, Gordon Conference 2012].

### 2.6. Interfibrillar Lymphatic Capillaries

Tiny tubular lymphatics (Φ ≤ 15 μm), presumably the initial lymphatic capillaries, were sparsely arranged in the diaphragmatic interstitial space with their major axis parallel to that of either the tendinous or muscular fibres (Figure 2). The density of the small lymphatic capillaries was higher in PL-M and PE-M than in PL-T and PE-T, respectively, while this distribution was opposite for larger lymphatic vessels. 

### 2.7. Distribution of Lymphatic Network within the Diaphragmatic Cross Section

In cross section, submesothelial lacunae were observed on both diaphragmatic sides, although PE side lacunae were significantly deeper compared to the PL side. Lymphatic conduits about 200 µm long depart from the PL and PE submesothelial structures (Figure 2) and penetrate through the muscular fibres up to the central portion of the diaphragm, with no difference between the side of origin. Deep, almost circular lymphatic vessels (≈60 µm radius) functioning as a collector system for the lymph to be carried out of the diaphragm through extra-diaphragmatic collectors were observed at a similar distance from both the PL and PE surfaces. Occasionally, the conduits from the PL and PE sides merge in the deepest diaphragm, forming transdiaphragmatic tunnels. 

## 3. Mechanisms Sustaining Lymphatic Function in the Diaphragm

The diaphragmatic lymphatic network is rather unique from several standpoints: (a) it must simultaneously absorb fluid from compartments characterized by quite different hydraulic pressures regimes, making fluid pressures much more subatmospheric in the pleural than in the peritoneal cavity or in diaphragmatic interstitium [13,18,19,20,21,22]; (b) unlike what is observed in most other tissues, in pleural and peritoneal cavities and in diaphragmatic tissue and lymphatics, hydraulic pressure widely oscillates in association to the respiratory and cardiac cycles [17,23,24]; (c) the lymph transport required to maintain fluid homeostasis in pleural and peritoneal cavities is quite different, with the turnover rate much higher in the latter [22,25]; (d) depending upon their location in the diaphragm (pleural or peritoneal, tendinous or muscular, superficial or deep), lymphatics display different morphological features and are exposed to variable and quite different mechanical stresses. Each of these factors must be taken into consideration when analysing diaphragmatic lymph formation in either normal or pathophysiological altered conditions. 

The formation of diaphragmatic lymph occurs when fluid passively enters the lymphatic structures driven by a net hydraulic pressure gradient (ΔP_lymph-net_) which develops when intraluminal lymphatic pressure (P_lymph_) falls below the pressure in the pleural, peritoneal or surrounding interstitial fluid. Direct measurements show that P_lymph_ values as low as ≈−20 cmH_2_O can be attained in the pleural apposition zone of the rodent diaphragm, both in open thorax [24] and during spontaneous breathing [13], clearly demonstrating the existence of ΔP_lymph_ values able to sustain drainage of pleural fluid. Data recorded from intercostal lymphatics in spontaneously breathing rats [21] show that ΔP_lymph_ may range between ≈−0.5 cmH_2_O at end-expiration and ≈−24 cmH_2_O at end-inspiration, indicating that pleural lymph may be produced throughout the whole respiratory cycle. 

P_lymph_ have never been measured in diaphragmatic lymphatics facing the peritoneal cavity. However, assuming a homogenous pressure regime in lymphatics of both diaphragmatic sides, and since peritoneal fluid pressure is higher (i.e., less negative) than pleural fluid pressure, ΔP_lymph_-promoting diaphragmatic peritoneal lymph flow is expected to be even higher than ΔP_lymph_-sustaining pleural lymph flow. 

Direct measurements of diaphragmatic P_lymph_ clearly present cyclic oscillations which reflect: (a) the transmission to the vessel lumen of stresses developed in the diaphragmatic tissue upon local cardiogenic [24] and/or respiratory [21] tissue displacements, a mechanism also defined as “extrinsic” and/or (b) the transmission of contractile waves along the lymphatic vessel wall, also known as “intrinsic mechanism”, well documented in non-thoracic lymphatic districts [11,12,26,27]. 

### 3.1. The Extrinsic Mechanism in Diaphragmatic Lymphatics

Even in open chest paralyzed rabbits, pleural diaphragmatic P_lymph_ shows cyclic pressure swings [13,16] able to sustain lymph formation due to cardiogenic displacements of the diaphragmatic tissue. During a cardiac cycle, myocardial contraction/relaxation alternance affects diaphragmatic P_lymph_ through: (a) rhythmic displacement of mediastinum and phrenic centre; (b) transmission of sphygmic waves along the aorta and the arteries feeding the diaphragm; (c) transmission of arterial pulsation though the hepatic surface facing the peritoneal diaphragmatic dome. Consequently, P_lymph_ is negative both during the diastolic and systolic cardiogenic phase (45%) or only during the diastolic phase (50%). The cardiac cycle also affects interstitial tissue pressure [24] that change either in phase or out of phase with respect to P_lymph_, determining cyclic expansions and compression of diaphragmatic lymphatics, thus contributing to development of ΔP_lymph_ and the formation of lymph from diaphragmatic interstitium. 

The role of the respiratory cycle in sustaining diaphragmatic ΔP_lymph_ is not directly known, as it is not possible to measure P_lymph_ in a closed chest during spontaneous breathing. However, in analogy with what has been observed in pleural intercostal lymphatics, where ΔP_lymph_ could instead be measured in intact rats during spontaneous or assisted ventilation [21], we hypothesise that the efficiency of the respiratory diaphragmatic contraction/relaxation cycle is more significant compared to that of cardiogenic oscillations. 

The impact of diaphragmatic tissue displacement on the lymphatic circuit critically depends upon the mechanical properties of the lymphatic vessel wall and of the surrounding muscular tissue. 

### 3.2. Diaphragmatic Tissue Mechanical Properties and Lymphatic Function

The mechanical features of the various portions of the diaphragmatic tissue play a relevant role in enhancing and modulating lymph flow. Indeed, through a Finite Element Model [28] taking into account (a) the compliance of the layer made of the lymphatic endothelium, the parietal mesothelium and its submesothelial tissue, (b) the stiffness of the diaphragmatic tissue around the lymph vessel and (c) the percentage of compliant or stiff tissue delimiting the lymphatic lumen, it has been possible to identify three types of lymphatic vessels, each with different mechanical properties and functions: (A)The vessels laying over the muscular/tendinous plane immediately beneath the mesothelial surface and delimited mostly by mesothelium are the most compliant ones; they are exposed to the highest circumferential tensile stresses, particularly along the contact edges of the mesothelial wall with the stiff diaphragmatic tissue basement. From the mechanical standpoint, such disposition favours high chances of dilatation of discontinuities of the mesothelial surface, represented, for example, by stomata and/or primary valves [6,10], thus making these vessels perfectly fitted to absorb fluid from the pleural and peritoneal cavities.(B)The vessels laying deeper within the diaphragmatic thickness [6] are characterized by a much stiffer (up to two orders of magnitude) wall compliance, a feature that ensures a more homogeneous distribution and a more efficient transmission of the circumferential stresses over the entire vessel surface. Because of these properties, deep vessels are more efficient than submesothelial ones in exploiting tissue stresses and propel diaphragmatic lymph in the network.(C)The vessels partly covered by mesothelium, and partly immersed in the muscular/tendinous tissue, show a transitory intermediate behaviour, being involved in both lymph production and propulsion.

### 3.3. The Intrinsic Mechanism in Diaphragmatic Lymphatics

In almost still, non-thoracic issues, lymph formation and propulsion are driven by cyclic contractions of lymphatic muscular cells in the vessel wall [26,27,29]. Contractions/relaxation of adjacent lymphangions, synchronized with opening/closure of the unidirectional valves [11,12,30,31,32], sustain ΔP_lymph_ between the interstitium and the lymphatic lumen [33], promoting both lymph formation and propulsion. Most studies on the intrinsic, active behaviour of lymphatic vessels have been performed in mesenteric lymphatics or large collecting ducts [11,12,26,27,29,32,34,35]. Although rare, direct recordings of slow-paced, spontaneous P_lymph_ swings, attributable to spontaneous contraction of the vessel wall, have also been documented at the outer peripheral rim of the muscular pleural diaphragm [13]. Indeed, while diaphragmatic lymphatics in the tendon and the inner muscular diaphragm, as well as larger and deeper collectors, are essentially devoid of muscle cells [6], lymphatic loops at the outer peripheral border contain sparse, smooth muscle α-actin fibres [15,16,17,24] which confer to these vessels a complex and composite functional behaviour. In fact, while a tract may actively and spontaneously contract, the adjacent segment can be completely passive or can contract only after intraluminal distension [16], suggesting a role of local stretch receptors in controlling vessel wall contractility. 

Several studies have been carried out to explain the genesis of the spontaneous contractility observed in the lymphatic vessel wall. Electrophysiological data demonstrated that lymphatic muscle cell contraction may be triggered by: (a) small transient cell depolarizations, likely due to spontaneous transient inward currents [36,37,38,39]; (b) the opening of voltage-gated L-type Ca^2+^ channels, whose threshold potential is reached through a slow depolarizing inward currents (*I*_h_), similar to what is observed in the cardiac sinoatrial pacemaker cells [36,37,40]; (c) voltage-gated Na^+^ channels initiating the action potential and subsequent phasic contraction of the vessels [40]. However, tested voltage-gated Na^+^ channel inhibitors only partially abolish the spontaneous contraction, particularly in smaller mesenteric vessels. In addition, the ionic mechanism of the spontaneous depolarization moving the cell potential to reach the activation threshold is not explained by these mechanisms. Hence, spontaneous firing of lymphatic muscle cells might actually depend upon alternative mechanisms, such as the occurrence of an inward hyperpolarization activated current, which has been proposed to explain spontaneous contractility in mesenteric lymphatics [41]. 

Studies on diaphragmatic lymphatics revealed that the spontaneously depolarizing current occurs through hyperpolarization-activated cyclic nucleotide-gated (HCN) channels whose inhibition leads to either a complete or significant blockade of spontaneous contractions, indicating their direct involvement in spontaneous lymphatic pacemaking [42]. HCN are voltage-gated ion channels permeable to Na^+^ and K^+^ ions [43,44] and triggered at membrane potentials of about −40/−60 mV [45]. All four homologous HCN_1_–HCN_4_ of the channel subunits are expressed in the wall of spontaneously contracting diaphragmatic peripheral lymphatics [42], even though HCN_2_ is the prevalent isoform and seems to be necessary to organize a functional channel; the HCN heteromeric channel isoforms seem unevenly dispersed in specific points of the lymphatic muscle, likely explaining the distinct behaviour of segments of the same diaphragmatic lymphatic loop where completely passive zones co-exist with strongly spontaneously contracting ones [16]. Channels of the HCN family have been proven to carry the inward “funny” current (I_f_), which induces the spontaneous depolarization of cardiomyocytes of the sinoatrial node [45,46,47] and plays a critical role in the automatic generation of action potentials in the heart. 

Lymphatic intrinsic contraction frequency and strength can be modulated by several factors, such as shear stress [24,48], transmural pressure gradients [21], low density lipoproteins [49] and Cav1.2 channels [50]. In spontaneously active lymphatic diaphragmatic lymphatics, the contraction frequency has been shown to be also sensible to variations in osmolarity of the surrounding interstitium [51]. Indeed, after an acute response which depends upon whether osmolarity is decreased or increased, contraction frequency and lymph flow invariably decrease in chronic alterations of interstitial osmolarity, with complete arrest in case of an alteration of extracellular sodium and/or chloride concentration. The increased contraction frequency observed immediately after exposure to hypo-osmotic solutions and its subsequent decrease can both be explained by the involvement of Volume Regulated Anion Channels (VRACs, [52]. Indeed, while the initial VRACs opening can depolarize lymphatic muscle cells [53,54,55] and increase the speed of threshold crossing in the pace-making cells [56,57], a subsequent chloride efflux through VRACs, not compensated or overcome by the active cotransport of Cl^−^ ions through NKCC exchanger, potentially leads to hyperpolarization and decreased contraction frequency. Alternatively, hyperosmolarity is known to induce membrane hyperpolarization of vascular endothelial and smooth muscle cells [58], and reduce mesenteric lymph flow [59] by means of several mechanisms, including aquaporins [60], KATP mediated responses [61], Na^+^ pump inhibition [62] and inhibition of VRACs, which are also involved in isotonic conditions [53,63]. The response of lymphatic vessels, including the diaphragmatic ones, to osmolarity might be pivotal in shaping the water and solute transport from the intestine [64] and in determining the bioavailability of drugs administered subcutaneously [65]. 

Finally, although likely of minor importance in modulating lymph flow in the diaphragmatic tissue, whose temperature is steady and similar to that of the thermic core, it is worth noting that spontaneous diaphragmatic lymphatic contraction frequency is sensitive to temperature changes—the maximal thermal sensitivity being achieved at a temperature of ≈36.7 °C [66,67]. Thermal sensitivity is lower in dermal lymphatics [66] (≈32 °C), suggesting that each vessel population presents a maximal contraction frequency at the temperature that these vessels are expected to face in the corresponding tissue district. 

## 4. Diaphragmatic Lymph Kinetics

Once formed in initial lymphatics, diaphragmatic lymph is propelled along the lymphatic network to reach, mainly, the right lymphatic duct to eventually be emptied into the venous blood stream. The path followed by the newly formed lymph along the diaphragmatic lymphatic circuit is highly variable and depends upon factors such as the specific location of the vessel within the diaphragm (superficial or deep, medial or peripheral, proximal to pleural or peritoneal side) as well as upon whether the diaphragm is relaxed or contracted within the respiratory cycle. 

During tidal expiration, with an almost relaxed or low-tone diaphragm, lymph flow is supported by active spontaneous contraction of specific segments of lymphatic loops [16,42,68] or by cardiogenic swings. The great majority of these loops lay over the most external portion of the muscular diaphragm where, in the so called “apposition zone”, the diaphragm faces the inner surface of the caudal chest wall without interposition of pulmonary tissue. Considering that, during tidal breathing, this area is the least mobile of the entire respiratory system, the mostly peripheral location of the spontaneously contracting segments might serve to support lymph propulsion in diaphragmatic regions not directly exposed to significant local tissue stresses. In these loops, as well as in linear vessels, fluid kinetics can be particularly complex [15]. Indeed, due to the intraluminal unidirectional valves which behave like resistive low velocity gates, isolating the entire loop from or connecting it to the inlet and outlet linear vessels allows the fluid to either recirculate, oscillate back and forth without further propelling and/or suddenly flow forward after opening a previously occluded valve. Although their filling/emptying sequences are still unclear, loops very likely serve as reservoirs, collecting lymph derived from the pleural/peritoneal cavities to subsequently converge it to linear vessels in the tendinous medial portion and/or to transverse and collect ducts of the deeper diaphragm. 

An important role in lymph formation and kinetics is played by inspiration, which implies contraction and change in the shape of the diaphragm with a shortening of the outer muscular fibres and an increased tensile stress of medial tendinous fibres. In an open chest rodent, the impact of diaphragmatic muscle contraction on lymph flow depended upon the alignment of a specific lymph vessel, with respect to muscle fibres orientation [15], as well as the depth of the vessel within the diaphragmatic tissue [28]. 

As far as vessel orientation is concerned, diaphragmatic muscle contraction simultaneously induces (a) compression of lymphatic vessels very close to and placed in roughly perpendicular direction with respect to muscle major axis, a phenomenon likely mirroring an increased transmural pressure across the vessel wall; (b) enlargement, shortening and an increased tortuosity of lymphatic vessels running parallel to the diaphragmatic muscle fibres [15]. The depth of the vessel within the diaphragm is also critical because that diaphragmatic muscle contraction causes an independent reduction of P_lymph_ in all superficial submesothelial vessels upon their orientation, with respect to muscle fibres; such P_lymph_ drop likely reflects an enlargement of the vessel diameter when the contracting underlying muscle shortens anisotropically, pulling on the vessel wall [23,33]. On the contrary, in deeper vessels, contraction of skeletal muscle fibres causes an orientation-dependent response; in fact, P_lymph_ decreased in vessels running parallel to the major muscle fibres axis and increased in vessels oriented perpendicular to the muscle fibres direction. 

In terms of diaphragmatic pleural fluid drainage, this quite complex behaviour likely guarantees absorption of pleural fluid into submesothelial lacunae, both during expiration, when the muscle fibres are almost relaxed, and during inspiration, when P_lymph_ in submesothelial diaphragmatic lymphatics decreases more than in the pleural fluid, thus increasing the pleural to submesothelial diaphragmatic lymphatics pressure gradients. In addition, on inspiration, the contracted diaphragmatic muscle fibres shorten, stretching the medial tendinous fibres, a condition that might squeeze deeper lymphatics, enhancing lymph propulsion. The propelling phase might be particularly efficient in deeper lymphatics whose low compliance [28] likely drops further on inspiration due to an increased elastance of the contracted diaphragmatic tissue [69]. Indeed, it has been shown [17,23,33] that most of lymph flow occurs during activation of the skeletal muscle, while spontaneous contraction of lymphatic muscle cells supports a modest but continuous propulsive action during skeletal muscle relaxation, thus guaranteeing a continuous lymph drainage from peritoneal and pleural cavities during the whole respiratory cycle. 

## 5. Role of Diaphragmatic Lymphatic in Pleural-Peritoneal Drainage under Physiological and Pathophysiological Conditions

Indirect estimates on various mammalian species suggest that physiologically normal pleural lymph flow amounts to 75% of the total pleural fluid turnover (about 0.2 mL/(kg·h) [70,71,72] or about 0.15 mL/(kg·h), 40% of which takes place through the pleural diaphragmatic network [73,74]. Hence, using the measurements of total diaphragmatic lymph flow available in sheep, ranging between 0.01 and 0.07 mL/(kg·h) [75,76], drainage through the pleural diaphragmatic lymphatics would account for ≈7 to 50% of the total diaphragmatic lymph flow; the remaining 50% would be derived from fluid drainage from the peritoneal cavity. A completely unveiled concern is whether (a) the pleural and peritoneal cavities may or may not be connected by passages through the diaphragm and (b) if a potential transdiaphragmatic path could play any role in physiological and/or pathological conditions. 

The confocal images presented in Figure 3 seem to provide evidence of a direct and structured pleural-peritoneal connection through the deep and central diaphragmatic lymphatic vessels, suggesting a freely accessible transdiaphragmatic lymphatic route [personal communication, Gordon Conference 2012]. 

An important issue to be addressed is whether, in spite of the observed morphology, fluid and solutes may actually cross the diaphragm under normal or pathophysiological conditions using such a pleuro-peritoneal route [77]. In physiological conditions, subdiaphragmatic peritoneal fluid pressure is higher than supradiaphragmatic pleural fluid pressure [20,78], suggesting that a favourable peritoneal to pleural transdiaphragmatic pressure gradient does exist during the whole respiratory cycle. 

However, a series of experiments aimed at detecting the transdiaphragmatic distribution of fluorescent dextrans, injected in either the pleural or peritoneal cavity [77], showed that, in physiological conditions, diaphragmatic lymphatics drain fluid and solutes from both the pleural and peritoneal cavities, preventing any migration of dextrans between the two cavities in the meantime. Indeed, independently of the injection site (pleural or peritoneal cavity), none of the injected dextrans crossed the whole diaphragm and only <0.5% reached the deeper central collectors; the great majority of dextrans remained in the lymphatics facing the diaphragmatic cavity of injection. When ascites was induced by an injection of saline enriched with high molecular weight dextrans [78], despite the increased transdiaphragmatic pressure gradient and a more homogenous dextran invasion in the diaphragmatic lymphatic network, only a negligible amount (0.17%) of the tracer reached the subpleural diaphragmatic lymphatic network. Similarly, experimentally induced intrapleural effusion was not followed by a transfer of dextrans to the peritoneal submesothelial lymphatic structures. Hence, data indicate that, in small and moderate pleural or peritoneal effusions, an increased fluid and solute drainage is achieved not only through a passive increase of the driving pressure gradients, but also though a more consistent recruitment of the tendinous lymphatics. 

Such results suggest that (a) diaphragmatic lymph flows down the pressure gradients arising between the pleural or peritoneal cavities and the lumen of diaphragmatic lymphatics draining from the respective serosal cavity [79]. Such gradients significantly exceed the transdiaphragmatic pressure gradients. In fact, P_lymph_ is very low in diaphragmatic lymphatics facing the pleural (about −10 mmHg [13,24]) and the peritoneal (−5 mmHg [25]) cavities, respectively. For this reason, fluid and solutes do not seem to cross the diaphragm, but are rather driven towards the deeper collecting lymphatics at the centre of the diaphragm to be subsequently drained out of the diaphragmatic network; (b) such an efficient compartmentalization requires that unidirectional intraluminal valves still display their normal opening/closing behaviour and are able to efficiently direct the drained fluid through the network and into the deeper vessels from which the lymph leaves the diaphragm. 

Based on these premises, potential pleuro-peritoneal leaks might indeed occur in conditions which imply pathophysiological alterations of the closing/opening capacity of intraluminal valves. Such conditions might be associated with very large pleural effusion and/or ascites when lymphatic structures might become overdistended, impeding the valve leaflets to efficiently close and thus, compromising the whole diaphragmatic lymphatic function. These considerations might explain the clinically relevant pleural effusions observed in patients undergoing peritoneal dialysis [80,81,82,83,84,85,86,87]. In these cases, diaphragmatic lymphatics likely lose their ability to sustain subatmospheric pressures in their lumen and the transdiaphragmatic pleuro-peritoneal pressure gradient becomes prevailing in driving fluid and solute fluxes. Furthermore, it has been shown that inflammation triggers a significant diaphragmatic lymphangiogenesis [8,9], suggesting an active proliferating response of lymphatic vasculature in order to overcome a likely impaired or insufficient function. 

Hence, although the complexity of a diaphragmatic lymphatic network has been increasingly revealed in the last decades by the continuous development of immunohistochemical and imaging techniques, many aspects of the mechanisms sustaining lymphatic drainage are still unclear and further research is needed on topics such as: (a) the actual correspondence between imaging/morphological evidences and functional dynamic behavior or, as an example, between observed increased lymphangiogenesis and measured increased lymph flow; (b) the interplay between tissue forces and active contraction in sustaining lymph flow; (c) the mechanisms through which lymph vessels “sense” the need for an increased lymph flow and trigger an increased draining function, either through lymphangiogenesis and/or through increased contraction frequency and stroke volume.

Clarifying these points will not only deepen our knowledge on the lymphatic vascular system in general, but will allow clinicians to better understand, and likely exploit, the role of the diaphragmatic lymphatic system in processes such as dialysis, tumor metastasis, infection, immunity and nutrition.

## 6. Conclusions

The lymphatic network of the mammalian diaphragm is morphologically designed to maximally exploit the anatomic and mechanical properties of the surrounding tissue to maintain the physiological fluid homeostasis in the pleural and peritoneal serosal cavity. Despite the thinness of the diaphragmatic tissue, particularly in its tendinous central area, the three-dimensional structure of the diaphragmatic lymphatic network modifies on the pleural with respect to the peritoneal side, and within the same diaphragmatic side, and on the muscular with respect to the tendinous regions of the diaphragm. Diaphragmatic lymphatic function is propelled and driven by a complex combination of spontaneously contracting lymphatic segments at the peripheral diaphragmatic rim and passive conduits exposed to swinging mechanical tissue stresses associated to cardiac and, more so, respiratory activity. Unlike what is observed in most other tissues, the impact of external tissue forces, in particular related to respiratory diaphragmatic contraction/relaxation cycles, prevails over the active, spontaneous contraction of lymphatic muscle in sustaining lymph flow. However, the latter seems to play a pivotal role in organizing the entire diaphragmatic lymph flow and in modulating its entity in the face of changes in osmotic or thermic tissue environment, thereby witnessing an unexpected complexity of the lymphatic vascular system.

## Figures and Tables

**Figure 1 biology-11-01803-f001:**
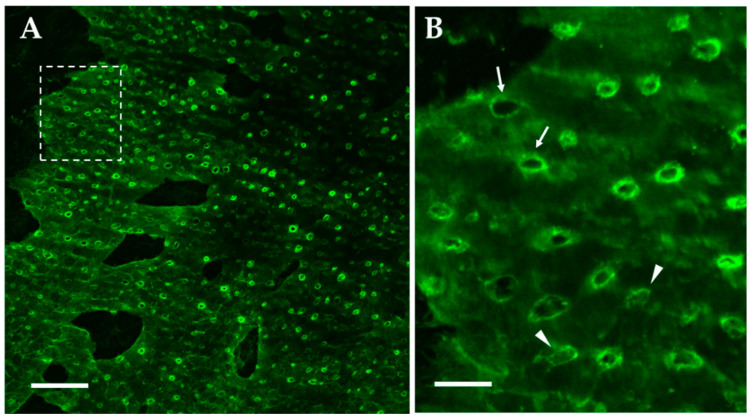
(**A**) Confocal image of a mouse diaphragmatic submesothelial lymphatic lacuna immunohistochemically stained with the fluorescent lymphatic vessel endothelial hyaluronan receptor (LYVE-1), a lymphatic endothelial marker. Several circular stomata are observable covering the entire surface of the lacuna. Bar: 50 µm. (**B**) Higher magnification of the lacuna surface delimited by the dashed line on Panel A. Stomata may show either a completely open (thin arrows) or closed (arrowheads) conformation. Bar: 20 µm.

**Figure 2 biology-11-01803-f002:**
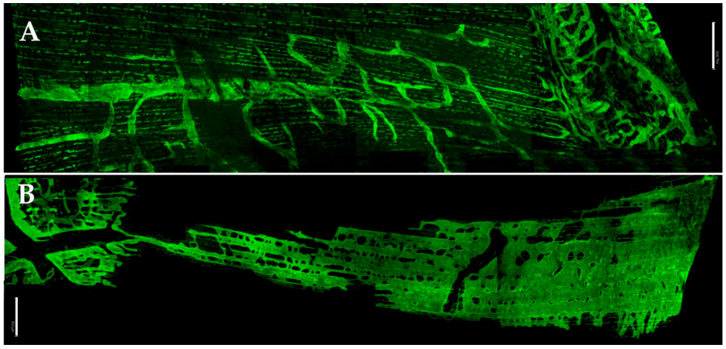
Confocal images of LYVE 1 stained mouse submesothelial lymphatic lacunae over the pleural (Panel (**A**)) and peritoneal (Panel (**B**)) sides of the diaphragm. A complex mesh of lymphatic vessels formed by confluent loops, linear vessels of variable sides equipped with lymphangions and thinner interfibrillar vessels are visible only in the pleural lymphatic side (**A**). On the peritoneal side, lymphatics are instead organized in large, flattened lacy-ladder shaped membranous structures (LLS). Bars: 300 µm.

**Figure 3 biology-11-01803-f003:**
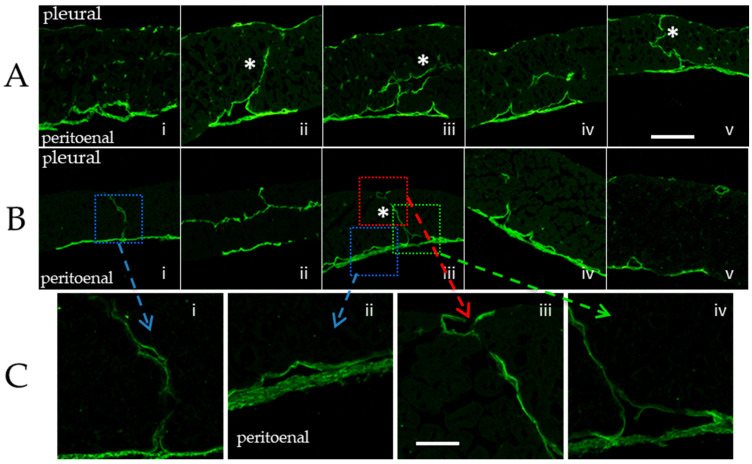
Panels (**A**,**B**). Low magnification confocal images of LYVE 1 stained mouse diaphragm. The pleural and peritoneal sides of the diaphragm (top and the bottom of panels (**A**) i–v may be connected by pleuro-peritoneal lymphatic vessels (asterisks). A different mouse cross section is shown in the image sequence (**B**) i–iv whose details are enlarged in panels (**C**) i–iv. Bar: 150 µm. Bar: 100 µm.

**Table 1 biology-11-01803-t001:** Average density, total stomata area (expressed as % of total area of submesothelial lacunae) and stomata radius in diaphragmatic lymphatics. PL-T: pleural-tendinous; PL-M: pleural-muscular; PE-T: peritoneal tendinous; PE-M: peritoneal-muscular. Data are presented as mean ± SD. Number of observation in brackets.

	Density n/mm^2^ of Lacuna	Stomata Area % of Lacuna Surface	Radius µm		
			Min	Max	Mean
**PL-T** (*n* = 32)	456.4 ± 104.9	1.9 ± 1.4	2.6 ± 0.15	4.9 ± 0.4	3.6 ± 0.2
**PL-M** (*n* = 37)	431.1 ± 150.1	0.4 ± 0.12	1.4 ± 0.05	1.9 ± 0.07	1.7 ± 0.05
**PE-T** (*n* = 604)	2669.4 ± 684.6	3.6 ± 0.65	1.5 ± 0.04	2.4 ± 0.04	1.9 ± 0.04
**PE-M** (*n* = 2827)	1893.0 ± 298	3.8 ± 0.85	1.9 ± 0.07	2.7 ± 0.02	2.3 ± 0.02

## Data Availability

All referenced data and reported results can be found in a publicly accessible repository.

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
