# Peer review of "Morphological, Mechanical and Hydrodynamic Aspects of Diaphragmatic Lymphatics"

_biology, 2022, doi:10.3390/biology11121803_

Round 1
Reviewer 1 Report
This is an excellent paper written by an expert in the field. I have no suggestions for improvements.
Author Response
I am grateful to Reviewer # 1 for his/her kind comments.
I have revised the manuscript taking into consideration the suggestion for a minor spell check. In addition, as required by Reviewer #2, I have added:
- references to few recent studies on diaphragmatic lymphatics, so references numbers and list have been modified
- a short perspective sentence on the need of further research directions
I hope these corrections will satisly Reviewers' requirements and enable the manuscript to be published in the Special Issue of Biology.
Best regards
Daniela Negrini
Reviewer 2 Report
The review nicely summarizes previous studies and layout diverse aspects of diaphragmatic lymphatics. The organ-specific lymphatics are of great interest as more studies are revealing their unique function besides the common features that lymphatics share. A perspective on the need for future research directions on diaphragmatic lymphatics would be an attractive section for the readers.
Figures: Co-immunostaining with other lymphatic markers such as Prox1 or Vegfr3 or Pdpn is highly recommended to clarify the nature of the labeled vessels and also to distinguish them from background staining.
In addition, including a number of recent studies on diaphragmatic lymphatics is recommended.
Author Response
I thank Reviewer # 2 for his/her constructive comments.
I have carefully revised the whole manuscript taking into consideration the suggestion for a minor spell check. In addition, as required, I have added:
- references to few recent studies on diaphragmatic lymphatics, so references numbers and list have been modified
- a short perspective sentence on the need of further research directions
Concerning the Figures, if the Referee permits, I would prefer to keep the Figures as they appear in the present version of the manuscript. Indeed, in this review I wish to convey the concept of the complexity of the lymphatic system from the standpoint of its overall structure and of the interplay between diaphragmatic muscular and tendinous fibers in sustaining lymph flow. In the images shown in the manuscript, all green structures identify LIVE1+ structures and there is no significant background noise that might create confusion in interpreting the significance of the structures themselves.
I hope these answer and the corrections made will satisly the Reviewer's points and enable the manuscript to be published in the Special Issue of Biology.
Thank you again and best regards
Daniela Negrini